# Long-Term Cumulative Effects of Repeated Concussions in Cyclists: A Neurophysiological and Sensorimotor Study

**DOI:** 10.3390/jfmk10040414

**Published:** 2025-10-22

**Authors:** Alan J. Pearce, Doug King

**Affiliations:** 1Swinburne Neuroimaging Facility (SNI), School of Health Science, Swinburne University of Technology, Melbourne, VIC 3122, Australia; 2Sports Performance Research Institute New Zealand (SPRINZ), Faculty of Health and Environmental Science, Auckland University of Technology, Auckland 11357, New Zealand; dking@aut.ac.nz; 3Auckland Bioengineering Institute, The University of Auckland, Auckland 1142, New Zealand; 4Wolfson Research Institute for Health and Wellbeing, Department of Sport and Exercise Sciences, Durham University, Durham DH1 3LE, UK

**Keywords:** sports-related concussion, road cycling, BMX, MTB, neurophysiology, transcranial magnetic stimulation

## Abstract

**Objectives**: Sports-related concussion (SRC) is mostly associated with contact and combat sports. However, emerging evidence suggest that cyclists are also at risk of repeated concussion injury. Moreover, long-term neurophysiological outcomes in cycling cohorts remain underexplored. This novel study investigated the long-term effect of repetitive concussions in cyclists. Road, mountain biking (MTB), and BMX riders with a history of concussions and self-reported persistent symptoms were assess for neurophysiology and cognitive–motor performance compared to previously concussed cyclists with no ongoing symptoms. Both groups were compared to age-matched with controls. **Methods**: Using a cross-sectional between-group design, 25 cyclists with a history of concussions (15 symptomatic, 10 asymptomatic) and 20 controls completed symptom reporting, cognitive and balance assessments (SCAT5), sensorimotor testing using vibrotactile stimulation, and neurophysiological assessments via transcranial magnetic stimulation (TMS). **Results**: Symptomatic cyclists reported a higher number of concussions compared to asymptomatic cyclists (*p* = 0.041). Cognitive testing revealed large effects (*d* > 1.0), with impaired concentration in symptomatic cyclists compared to controls (*p* = 0.005). Motor assessments demonstrated large effects (*d* > 1.0), with slower tandem gait times (*p* < 0.001) and greater errors (*p* = 0.02) in the symptomatic group. Sensorimotor testing indicated slowed simple reaction times (*p* = 0.001) and poorer temporal order judgement *(p* = 0.038). TMS showed large effects (*d* > 1.0) in increased cortical inhibition in the symptomatic group, with prolong cortical silent periods (*p* < 0.05) and large effects (*d* > 1.0), and reduced short interval intracortical inhibition (*p* = 0.001) compared to asymptomatic cyclists and controls. **Conclusions**: Cyclists reporting persistent symptoms showed greater cortical inhibition and impaired cognitive–motor performance, consistent with findings in contact sport athletes. These results suggest that repeated concussions in cycling carry risk of chronic neurophysiological alterations. Cycling disciplines should consider more rigorous concussion identification protocols and stricter management strategies to mitigate persistent and long-term consequences.

## 1. Introduction

Sports-related concussion (SRC) and repetitive head impacts (RHIs) continue to be a concern across many sports worldwide, particularly in combat and contact sports. Risks of long-term outcomes such as cognitive impairments and neurodegenerative disease have been described in the mainstream media as an “existential threat” to sports [1,2].

Athletes experiencing repetitive SRC are at risk beyond clinical outcomes, which contributes to public health and socioeconomic burdens. Those who experience repeated concussions are at increased risk of prolonged recovery, persistent neurological symptoms, and premature career retirement, leading to a potential loss of income and reduced occupational opportunities [3,4]. Further, the economic costs associated with concussion include direct medical expenses, as well as indirect costs involving rehabilitation, loss of productivity, and long-term care needs [5].

In limited or no-contact sport, however, the rates of SRC are lower. Still, the risk of recurrent concussion injury is still apparent and has prompted many organisations to adopt or update their concussion policies and procedures for all levels of sport participation [6,7]. Cycling fits into this category, with Swart et al. [8] suggesting that SRC accounts for between 1.3% and 9.1% of all cycling-specific injuries (lower than the football codes).

A more recent epidemiological analysis, completed in 2024, showed that road cycling, bicycle motorcross (BMX), and mountain biking (MTB) had reported concussion in 4.68%, 6.03%, and 5.86% of all injuries, respectively [9]. However, exact numbers are difficult to determine as, like many other sports, SRC is underreported or not reported by cyclists [10]. Reasons for this are varied and include lack of knowledge/awareness [10,11], sex [12], and a cultural belief that concussion are “not serious” [10], with teammate peer pressures to downplay concussion by encouraging hiding of or ignoring any concussion-type symptoms [13].

On a more positive note, cycling is addressing the issues of SRC with the publication of cycling-specific concussion assessments. For example, the Harrogate consensus agreement [8], roadside concussion assessment for cycling [14], and the emergence of concussion and brain trauma research across cycling sub-disciplines [15,16].

Research to date has focused mostly on the incidence of SRC. However, little is known on the longitudinal aspects of repetitive concussion in cycling. While anecdotal accounts of long-term cognitive impairments [17] and concerns regarding neurodegenerative disease [18] have been reported in competitive track cycling and BMX, respectively, to date there has been no systematic study of longitudinal outcomes following concussion in cyclists.

Previous studies investigating longitudinal effects of repeated concussions across contact and combat sports and have examined the pathophysiology of the corticospinal pathways [4,19,20,21,22]. Indeed, it has been argued that changes in the motor pathways are typically a sign of clinical manifestations of repetitive chronic brain injuries in athletes [23].

A non-invasive technique that can sensitively measure cortical inhibitory and excitatory mechanisms of the corticospinal pathway is single-pulse transcranial magnetic stimulation (TMS). Used extensively in clinical and experimental neurology [24,25,26], TMS provides robust measures of the excitability of the corticomotor pathways via the amplitude of the motor evoked potential (MEP), as well as cortical inhibition reflecting γ-aminobutryic (GABA) receptor activity through cortical silent period (cSP) duration with single-pulse TMS. Paired-pulse TMS also allows for the investigation of GABAertic activity through short-interval intracortical inhibition (SICI), where interstimulus intervals between 1 and 5 ms, quantifying GABA_A_, and long-interval intracortical inhibition (LICI) presented at an interstimulus interval between 50 and 200 ms reflect GABA_B_ activity. While cSP and LICI reflect GABA_B_ receptor activity, these two parameters reflect different neural and circuit mechanisms [27]. TMS has been used extensively in SRC research, showing increased cortical inhibition following concussions and in those with persistent symptoms [28].

Complimenting TMS, sensorimotor testing aims to non-invasively assess central nervous system function through tactile sensory perception. Through delivering precise vibrotactile stimuli to the fingertips (specifically digits D2 and D3), testing can evaluate cortical processing mechanisms based upon stimuli frequency, duration, timing, and amplitude [29]. Previous studies using sensorimotor assessment have supported TMS findings in slowed reaction time and slowed responses rates across various cognitive–motor testing in athletes with a history of concussions compared to age-matched controls [20].

Rabadi and Jordan’s hypothesis have subsequently been confirmed in studies using TMS in contact and combat athletes [19,21,30]. However, TMS studies investigating concussion have not yet been undertaken in cyclists. With increased concerns on not only high rates of traumatic injury but also risks of repetitive non-concussive brain trauma, particularly in off-road [15,16] and alternative cycling events (e.g., freestyle BMX), studies into longitudinal outcomes following concussion is required.

The present study evaluated sensorimotor performance and neurophysiological excitability and inhibition in a cycling mixed cohort with and without persistent symptoms. It was hypothesised that cyclists with persistent symptoms would report a history of a greater number of repeated concussions, poorer sensorimotor test performance, and greater motor system inhibition.

## 2. Materials and Methods

This study utilised a between-group design, with participants visiting the laboratory for a single visit. Twenty-five cyclists (21 males, 4 females) with a history of concussions and twenty age-matched controls (17 males, 3 females) participated in testing. Controls had no history of concussions or participation in contact sports, an absence of self-reported cognitive or behavioural concerns, nor diagnosed neurological impairments or psychiatric disorders. Table 1 outlines all participants’ demographics and concussion history while Figure 1 presents a simple schematic showing participant recruitment. An a priori sample size calculation for ANOVA based on previous TMS data in concussion for determination of effect size f = 0.5, alpha = 0.05, power = 0.80 was a minimum sample size of 34. All participants provided written informed consent to participate in the study. The Institutional Review Board approved all study protocols (HEC18005, see full statement at end of this manuscript) conforming to the guidelines set out by the Declaration of Helsinki.

Cyclists were recruited via word of mouth throughout various cycling communities. Participants were from a range of disciplines including road (n = 7), mountain biking (MTB, n = 11), and bicycle motocross (BMX, n = 7). BMX riders included both racing and freestyle BMX. Based on self-assessment symptom reporting (Table 2), the cycling group was divided into those whose symptom reporting met the clinical cut-off of >10.5 (“symptomatic”: n = 15), compared to riders who were under this cut-off (“asymptomatic”: n = 10) [28]. Both groups were compared to age-matched controls [31]. Other than concussion history, cycling participants had no neurological or psychiatric diagnoses.

Prior to testing, participants completed pre-screening for suitability to TMS, the fatigue and related symptom survey [31], sensorimotor processing via vibrotactile stimulation [29], and TMS. Vibrotactile stimulation and TMS were conducted in a counterbalanced order to reduce potential stimulation serial effects [4].

### 2.1. Symptom Assessment

Symptoms were recorded on the Post Concussion Symptom Scale (PCSS) from the Sports Concussion Assessment Tool Version 5 (SCAT5) as number of self-reported symptoms (maximum 22 symptoms) and severity of each identified symptom (maximum 132 points) [32].

### 2.2. Cognitive and Motor Testing

Cognitive and motor testing were completed using elements of the Sport Concussion Assessment Tool V5 [32]. Immediate memory was assessed using the 10-word recall test involving three repeat trials and was scored as the total correct responses from three trials of 10 words (maximum score 30). Delayed memory testing was conducted approximately 15 min after the immediate testing when the participant had undertaken concentration and motor testing tasks. Participants were asked to recall how many of the words they were given.

Concentration was measured using the digit backwards assessment and reverse months of the year. Participants were given a set of numbers, from three digits to a maximum of six digits, and instructed to respond the opposite order presented to them. For this study, scoring for concentration involved both trials for each level of difficulty; therefore, scores were out of a maximum of eight rather than four. Participants were then asked to recall the months of the year in reverse order. Maximum score for concentration was nine (eight for digit backwards plus one for reverse months).

Motor testing involved the tandem gait, where participants were instructed to walk on a 10 m marked straight line with one foot in front of the other, with heel and toe touching of each foot, as quickly as possible without making any errors. Two attempts were provided with the fastest time without error recorded.

Balance was measured with single leg and tandem leg stance (Romberg test). Following familiarisation with eyes open, both tests were completed with eyes closed and assessed by the number of errors (i.e., using other leg to stop falling over) in each 30 s trial.

### 2.3. Sensorimotor Testing

Employing previously published protocols in concussion participants [4], sensorimotor assessment was completed via a portable vibrotactile stimulation device (Cortical Metrics, Chapel Hill, NC, USA). The device, similar in appearance to a standard computer mouse, contains two cylindrical probes (5 mm diameter) positioned at the top and towards the front of the device. These probes, driven by the computer via a USB cable, provided a light vibration stimulus at frequencies between 25 and 50 Hz sensed by the participant’s index (D2) and middle (D3) digits of their non-dominant hand [13].

The testing battery consisted of five distinct tasks: (1) simple reaction time, completed at the beginning and end of the battery to ascertain any fatigue from the challenge of the tasks; (2) choice reaction time by choosing which digit, D2 or D3, felt the stimulus pulse; (3) amplitude discrimination from deciding which digit, D2 or D3, felt a greater intensity during a trial of separate sequential vibration of the digits, followed by a trial of simultaneous vibration of the digits; (4) temporal order judgement by determining of the two stimulus pulses which digit, D2 or D3, felt the first stimulus pulse; and (5) duration discrimination where participants needed to decide which vibration stimuli presented simultaneously lasted for a longer time. For full detailed protocols, the reader is referred to Holden et al. [33] and Tommerdahl et al. [29].

### 2.4. Transcranial Magnetic Stimulation and Surface Electromyography

Using well-established methods [4,20,34,35], single- and paired-pulse TMS were applied over the area of the contralateral primary motor cortex (M1) targeting the participant’s first dorsal interosseous (FDI) muscle. For electromyography (EMG), surface electrodes (ADInstruments, Sydney, Australia) with an inter-electrode distance of 2 cm were placed over the FDI of the participant’s dominant hand following the recommendations of the surface electromyography for non-invasive assessment of muscles (SENIAM) project [36]. Signals were amplified (×1000), filtered (10–1000 Hz), and sampled at 2 kHz, recording 500 ms responses (100 ms pre-stimulus, 400 ms post-stimulus; PowerLab 4/35, ADInstruments, Sydney, Australia). All TMS procedures were undertaken with the TMS checklist for methodological quality [37].

A MagVenture R30 stimulator (MagVenture, Farum, Denmark) with a C-B60 butterfly coil (outer dimension 75 mm) was used to generate MEPs. During TMS testing, participants wore a fitted cap (EasyCap, Wörthsee, Germany), marked with sites at 1 cm spacing in a latitude–longitude configuration to ensure correct coil location. The cap was positioned with reference to the nasion-inion and interaural surface anatomical landmarks [4]. Prior to data collection, participant’s “optimal site” was identified, where the largest MEP was observed. The optimal site was then used throughout the TMS protocols.

#### 2.4.1. Single-Pulse TMS

Motor threshold (MT) is identified as the lowest stimulation to observe a MEP of at least 50 µV with the muscle at rest, or 200 µV during muscle contraction [38]. Active motor threshold (aMT) was identified during a controlled, low-level voluntary contraction (10 ± 3% maximal voluntary contraction, MVC, taken prior to TMS) of the FDI muscle. Stimulator output was increased in 5% steps, starting from 15% of the stimulator output, then 1% of stimulator output, to determine exact aMT, while the participant held the isometric contraction until discernible MEPs were observed [39,40].

Once aMT was identified, 20 stimuli (four sets of 5 stimuli) were delivered at 130%, 150%, and 170% of aMT. To avoid stimulus anticipation, stimuli were spaced 4–8 s apart with a 30 s rest between each set to reduce muscle fatigue.

Corticospinal latency was measured from the TMS stimulus artefact on the EMG to the onset of the MEP waveform. MEP waveform amplitude was calculated from measuring the peak-to-trough difference on the EMG. Duration of the cSP was measured from the initial onset of the MEP waveform to the return of the uninterrupted EMG. [34]

To address between-participant variability of the MEP [24], ratios were calculated by the duration of the cSP with the amplitude of the MEP [41,42]. The cSP:MEP ratio reflects the balance between inhibitory and excitatory mechanisms observed in the MEP waveform. SRC cohorts using cSP:MEP ratios have been previously published [43].

#### 2.4.2. Paired-Pulse TMS

Paired-pulse MEPs for SICI were measured with the FDI at a slight contraction of 5% MVC with a conditioning stimulus of 80% aMT and a test stimulus of 130% at an interstimulus interval of 3 ms [4,19]. Fifteen stimuli (in three sets of five) were delivered at random intervals between 8 and 10 s, with a 30 s interval after each set of five. SICI was expressed as a ratio of the paired-pulse MEP amplitude to the MEP amplitude measured at 130% AMT [4,19,44]. LICI was measured at an interstimulus interval of 100 ms with suprathreshold conditioning and test stimuli at 130% of aMT [4,19] and also expressed as a ratio of the conditioning stimulus and test stimulus [4,19,45].

### 2.5. Statistical Analyses

Data was screened for normality using Shapiro–Wilk tests and found to be normally distributed (*W* = 0.949–0.981; *p* > 0.05). ANOVA was utilised to test differences between groups for cognitive and motor testing elements of the SCAT5, sensorimotor testing, and TMS. Where ANOVA detected significance, post hoc comparisons using Tukey adjustments were employed. Exceptions to this were analysis of number of concussions, time since last concussion, and symptom reporting, which were compared between the cycling participants using an unpaired *t*-test. Where appropriate, Cohen’s *d* effect sizes were employed with 0.2 (small), 0.5 (moderate), and 0.8 (large), used to describe the magnitude of effects [46]. Alpha was set at 0.05. Data are presented as mean (±SD).

## 3. Results

All participants completed testing with no adverse effects. No differences were observed in age or the time since the last concussion experienced (Table 1). However, cyclists with ongoing symptoms reported a greater number of concussions (*p* = 0.041, *d* = 0.962; Table 1). Cyclists expressing ongoing concerns scored significantly higher on the fatigue and related symptom scale (*p* < 0.001), number of symptoms (*p* < 0.001), and severity of symptoms (*p* < 0.001; Table 2).

### 3.1. Cognitive and Motor Assessment

One-way ANOVA for cognitive testing showed no differences in immediate (F_(2,42)_ = 2.684, *p* = 0.081) or delayed memory recall (F_(2,42)_ = 2.803, *p* = 0.073; Table 3). However, concentration revealed a significant difference (F_(2,42)_ = 6.151, *p* = 0.005) with post hoc testing showing symptomatic cyclists performing worse in concentration (digit backwards) than control participants (*p* = 0.005; *d* = 1.087).

Motor tests (Table 4) revealed significant differences between groups for tandem gait walk time (F_(2,42)_ = 14.65, *p* < 0.001) and for number of errors of tandem leg stand (F_(2,42)_ = 4.33, *p =* 0.020). Post hoc testing for tandem gait walk time showed symptomatic riders were slower than both asymptomatic (*p* < 0.001, *d* = 2.258) and control groups (*p* < 0.001, *d* = 1.610). Tandem leg stance showed symptomatic cyclists had greater errors than both asymptomatic (*p* = 0.043, *d* = 0.862) and control groups (*p* = 0.041, *d* = 1.012). Conversely, no differences were found between groups for number of errors with single leg balance (F_(2,42)_ = 1.296, *p* = 0.288).

### 3.2. Sensorimotor Testing

Two-way ANOVA for repeated simple reaction time testing reported a significant group–time interaction effect (F_(2,42)_ = 20.1, *p* < 0.001) and main effects for groups (F_(2,42)_ = 20.1, *p* = 0.005) with post hoc testing showing significant slowing in reaction time between symptomatic cyclists compared to control participants (*p* = 0.001; Figure 2). No interaction effects were found with reaction time variability (*p* = 0.359) or main effects for time (*p* = 0.465) or between groups (*p* = 0.465). Table 5 presents other variables showing only a significant difference between groups for temporal order judgement (F_(2,42)_ = 4.2, *p* = 0.022), with post hoc testing showing symptomatic riders performed significantly worse than both asymptomatic and controls (*p* = 0.038).

### 3.3. Transcranial Magnetic Stimulation

Table 6 shows data from TMS. No differences were observed between groups for aMT (F_(2,42)_ = 0.25, *p* = 0.776) or corticospinal latency (F_(2,42)_ = 1.90, *p* = 0.182). While no differences between groups was found in cSP:MEP ratio at 130%aMT (F_(2,42)_ = 1.29, *p* = 0.285), significant differences were found at 150%aMT (F_(2,42)_ = 4.29, *p* = 0.020) and 170%aMT (F_(2,42)_ = 3.62, *p* = 0.035). Post hoc testing at 150%aMT showed differences between symptomatic and control groups (*p* = 0.018; *d* = 1.265). Similarly, differences at 170%aMT were found between symptomatic and controls (*p* = 0.046; *d* = 1.040). Figure 3 presents single-pulse MEP waveforms illustrating differences in cortical inhibition.

SICI ratio (Table 6, Figure 4) showed significant differences between groups (F_(2,42)_ = 9.39, *p* = 0.001), with post hoc testing revealing significant differences between symptomatic versus asymptomatic (*p* = 0.001; *d* = 1.319) and control groups (*p* = 0.013; *d* = 2.742). Although large effects were observed between the symptomatic and asymptomatic groups (*d* = 0.89), and controls (*d* = 0.85), no significant differences were found with LICI ratio (F_(2,42)_ = 2.41, *p* = 0.112; Figure 5).

## 4. Discussion

The primary objective of this study was to quantify the excitability of the corticomotor pathways to understand cyclists with self-reported persistent issues compared to those who had reported no ongoing issues and age-matched controls. Results from this study suggest that multiple SRCs result in chronic motor control dysfunction, with TMS data support from intracortical inhibitory system abnormalities. Specifically, cyclists with persistent symptoms showed the following: (1) a greater number of concussions and higher self-reported F&RSS scores; (2) poorer concentration, slowed reaction times, and timing judgement; (3) increased errors in motor tasks including tandem gait and tandem leg stance balance; and (4) increased cortical inhibition from TMS.

While these findings have been previously reported in contact sport athletes [21,22], to the best of the authors’ knowledge, this is the first reported data in cyclists. It should be noted, however, that while “symptomatic” cyclists showed notably poorer outcomes than “asymptomatic” cyclists and controls, overall, these results, with the exception of 10-word recall, were not outside of previously published clinical reference limits in multi-sport athlete studies [47,48].

Conversely, sensorimotor testing showed meaningfully slowed reaction times and timing judgement impairments. While not statistically significant, reaction time differences between the symptomatic and controls (~50 ms) could have an increased risk of further injury. For example, in studies of hazard detection–response times in the range of 390–600 ms, an additional response time of 50 ms (8–13% increase) is not trivial, particularly when a rider may need to avoid a sudden obstacle [49]. While the differences in reaction time have been reported across athlete groups [20,43], the finding of temporal order judgement is interesting, as Tommerdahl and colleagues have shown this measure reflects frontal–striatal cortex processing with reportedly poorer sequencing and coordination associated with poorer performance on this metric [29]. Further, cortical inhibition was seen as a prolonged cortical silent period and decreased SICI, suggesting that the cortical pathways remain affected and may explain impaired reaction times, timing performance, and tandem gait and balance performance. While LICI was not meaningfully different in the “symptomatic” group, the large effect size does support LICI differences reported in previous research [4,19]. Taken together, these findings suggest that a history of concussions induce ongoing alterations of intracortical inhibition mediated by GABA_A_ and GABA_B_ receptor activity [4,21,22].

Taken together, these results suggest that ongoing increased cortical inhibition and cognitive–motor performance decrements could put cyclists at risk of further injury. Indeed, the group with reported ongoing concerns (“symptomatic”) showed they have a greater history of concussion injuries. However, given the scope of the study, there was no data taken on overall injury history in the cyclists. Despite this limitation, there is strong evidence from multiple systematic reviews that show increased risk of concussions and musculoskeletal injuries in athletes with a history of multiple concussions [50,51,52,53], with research suggesting residual disturbance of cortical potentials in neuronal networks involved affecting postural movements and reducing the threshold for further brain re-injury [30].

### Strengths and Limitations of the Study

A strength of this novel study involves the use of neurophysiology and sensorimotor testing to investigate sub-clinical changes in those expressing ongoing concerns with regard to their concussion history. These methods have the potential to compliment other imaging and clinical assessments, showing the value of multi-modality testing for those with complex or persistent symptom concerns.

Limitations of this study include self-assessment reporting by the cyclists. While self-reporting does come with concerns generally [22], the aim here was to use a validated instrument for comparative purposes rather than quantifying their current situation. Of note, cyclists are well-known for down-playing their symptoms and also will not seek or even ignore medical advice to continue riding [10]. However, future research should aim to quantify cyclist’s medical histories, including non-concussive injuries that may impact self-reporting. A further limitation is that many in the symptomatic group were continuing to ride and may have affected the results. As discussed, many cyclists, particularly those who take greater risks with BMX jumping and MTB, do not heed medical advice and continue when they may be experiencing sub-concussive events [15,16]. However, we aimed to best control the cycling participants by excluding those who had a concussion within six months prior to testing. The study is also limited in that general musculoskeletal injury history was not able to be taken and could influence tandem gait and balance scores [54,55]. However, similar to concussion history, as part of the screening process, individuals were not eligible for participation if they had a musculoskeletal injury within the previous six months.

## 5. Conclusions

In conclusion, this novel study in cyclists shows that repetitive sports concussions induced persistent GABAergic changes as seen with increased cortical inhibition with TMS, along with balance changes, slowed reaction times, and impaired timing judgements. Further studies are required to elucidate the mechanisms and relationships of increased cortical inhibition on neuromuscular function, such as balance and reaction times as seen in this study. However, non-invasive techniques such as sensorimotor testing and TMS provide insights into central nervous system information processing mechanisms that can assist the clinical assessment for cyclists across different sub-disciplines. Cycling federations should consider physiological testing for cyclists who have a history of recurrent concussions and, beyond normal timelines of recovery, experience persistent symptoms.

## Figures and Tables

**Figure 1 jfmk-10-00414-f001:**
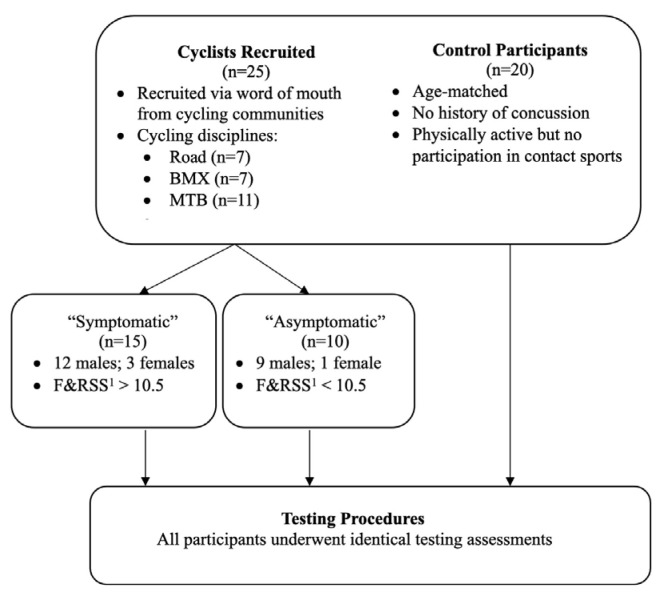
Simple schematic flow chart of participant recruitment and group allocation. ^1^ F&RSS = fatigue and related symptom scale [31].

**Figure 2 jfmk-10-00414-f002:**
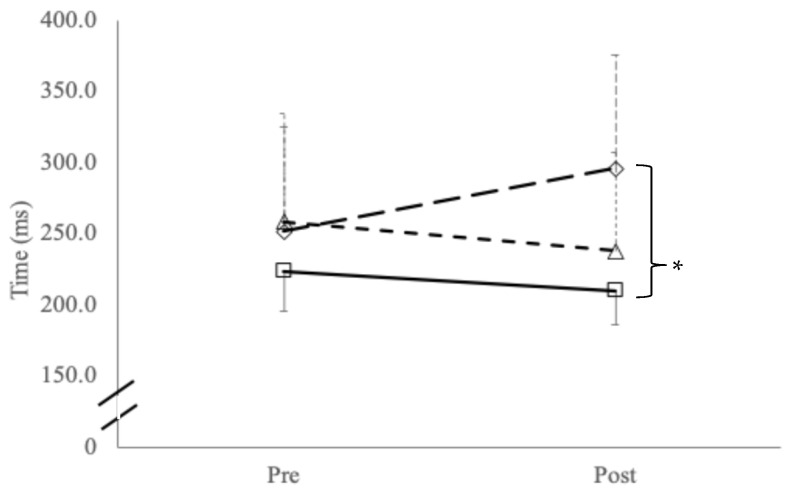
Simple reaction time (mean ± SD) between groups (long dash: symptomatic cyclists; short dash: asymptomatic cyclists; solid line: controls); * indicates significant difference between symptomatic cyclists versus controls (*p* = 0.001).

**Figure 3 jfmk-10-00414-f003:**
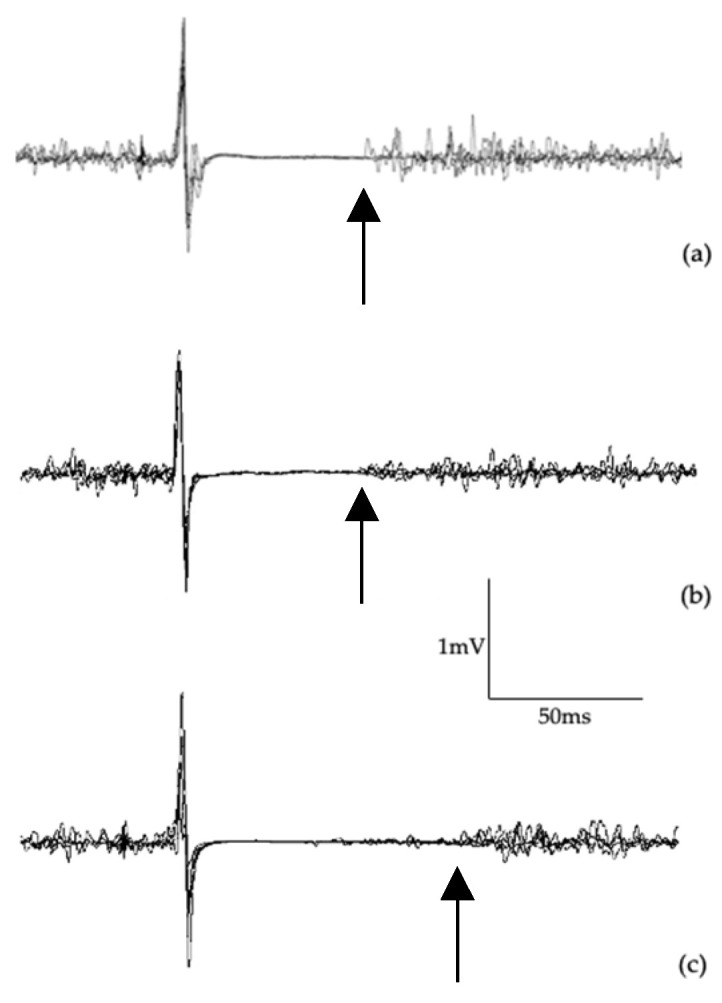
Example of five overlaid single-pulse MEPs at 170%aMT from a control participant (**a**), an asymptomatic cyclist (**b**), and a symptomatic cyclist (**c**). Arrow indicates end of cSP.

**Figure 4 jfmk-10-00414-f004:**
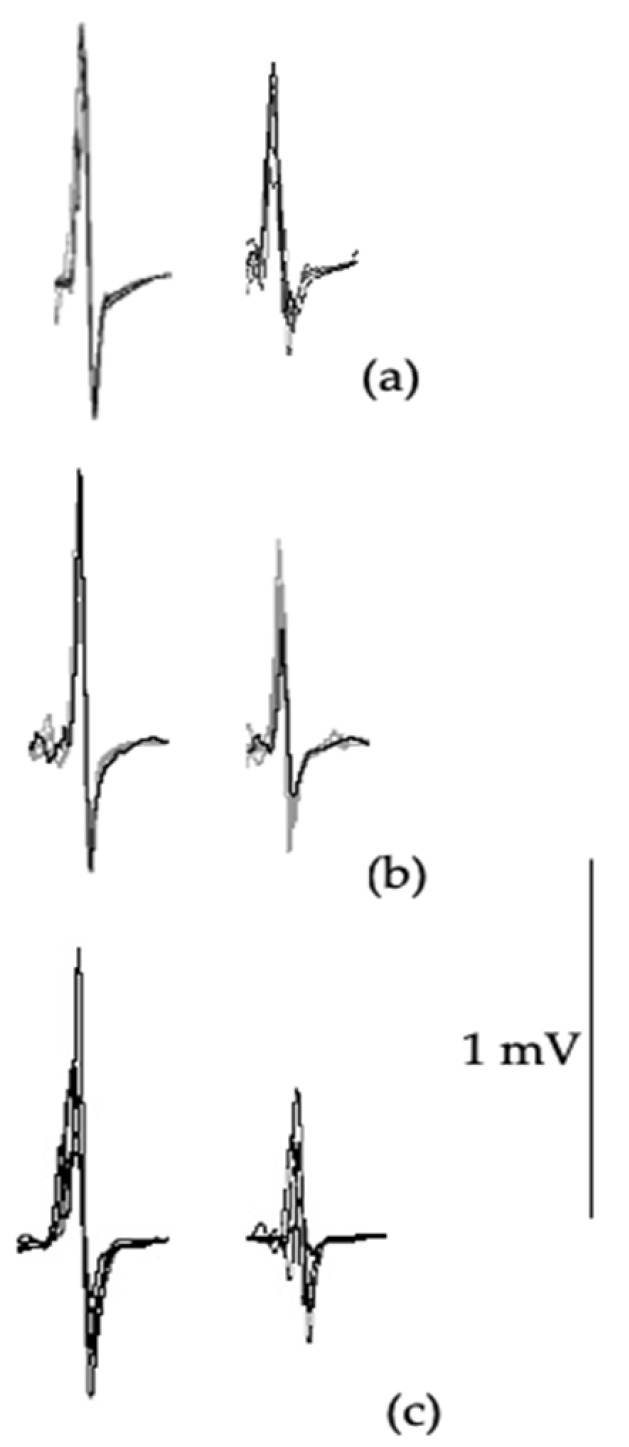
Example of control MEP (left waveforms) and SICI waveform (right waveforms) from 3 ms paired-pulse TMS (right waveforms) from a control participant (**a**), an asymptomatic cyclist (**b**), and a symptomatic cyclist (**c**). SICI waveform ratio is compared to their individuals own MEP waveform for calculation of the SICI ratio.

**Figure 5 jfmk-10-00414-f005:**
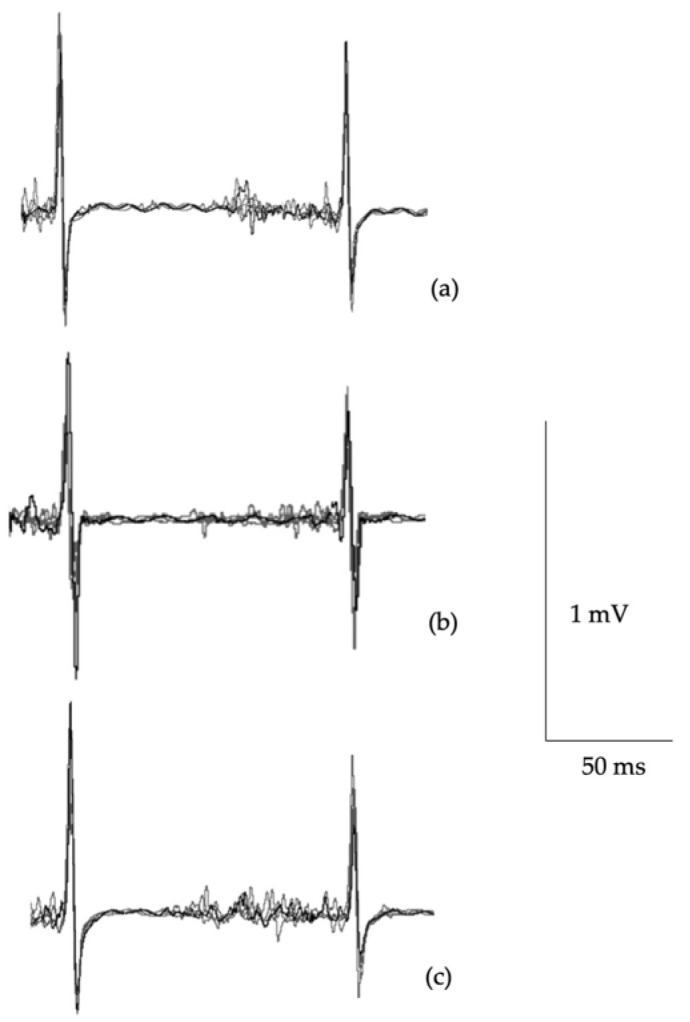
Example of LICI waveforms at 100 ms interstimulus interval paired-pulse TMS from a control participant (**a**), an asymptomatic cyclist (**b**), and a symptomatic cyclist (**c**). LICI test waveform amplitude (second waveform) is compared to the control waveform amplitude (first waveform) and expressed as a ratio.

**Table 1 jfmk-10-00414-t001:** Participant demographics and concussion history (number and time since last injury). Data reported in mean (±SD).

	Age (Years)	Number of Reported Concussions	Time Since Last Concussion (Years)
Symptomatic (n = 15; 12 m, 3 f)	29.3 (±4.6)	6.9 ^1^ (±5.1)	2.7 (±1.7)
Asymptomatic (n = 10; 9 m, 1 f)	26.7 (±3.6)	3.3 (±1.4)	1.5 (±0.8)
Controls (n = 20; 17 m; 3 f)	25.8 (±6.2)	n/a ^2^	n/a

^1^ Significant difference between symptomatic and asymptomatic groups (*p* = 0.041); ^2^ n/a = not applicable.

**Table 2 jfmk-10-00414-t002:** Participant symptom survey scores. Data reported in mean (±SD).

	Total F&RSS ^1^ (Max 44)	Reported Symptoms ^2^ (Max 22)	Severity of Symptoms ^2^ (Max 132)
Symptomatic (n = 15; 12 m, 3f)	18.2 ^3^ (±3.8)	10.7 ^3^ (±3.9)	23.5 ^3^ (±12.7)
Asymptomatic (n = 10; 9 m, 1 f)	6.2 (±3.4)	3.3 (±3.0)	4.4 (±4.1)
Controls (n = 20; 17 m; 3 f)	6.3 (±2.9)	3.3 (±3.2)	3.7 (±3.6)

^1^ F&RSS: fatigue and related symptom score [31]; ^2^ SCAT5: Sports Concussion Assessment Tool Version 5 [32]; ^3^ Significant difference between symptomatic versus asymptomatic (*p* < 0.001) and controls (*p* < 0.001).

**Table 3 jfmk-10-00414-t003:** Participant memory and concentration testing scores between groups. Data reported in mean (±SD).

	Immediate Recall (Max 30)	Delayed Recall (Max 10)	Concentration (Max 9)
Symptomatic (n = 15; 12 m, 3 f)	18.7 (±2.5)	5.9 (±1.0)	6.1 ^1^ (±1.7)
Asymptomatic (n = 10; 9 m, 1 f)	20.2 (±2.7)	6.9 (±0.7)	6.5 (±1.0)
Controls (n = 20; 17 m; 3 f)	20.7 (±2.4)	6.5 (±1.1)	7.7 (±1.2)

^1^ Significant difference between symptomatic and control groups (*p* = 0.005).

**Table 4 jfmk-10-00414-t004:** Participant tandem gait time and errors for balance testing between groups. Data reported in mean (±SD).

	Tandem Gait time (s)	Single Leg Stance—Eyes Closed (Errors)	Tandem Leg Stance—Eyes Closed (Errors)
Symptomatic (n = 15; 12 m, 3 f)	9.2 ^1^ (±1.2)	2.1 (±1.8)	1.3 ^2^ (±1.3)
Asymptomatic (n = 10; 9 m, 1 f)	6.6 (±1.1)	1.7 (±2.7)	0.4 (±0.7)
Controls (n = 20; 17 m; 3 f)	7.1 (±1.4)	1.1 (±0.9)	0.5 (±0.5)

^1^ Significant difference between symptomatic versus asymptomatic (*p* < 0.001) and controls (*p* < 0.001); ^2^ Significant differences between symptomatic versus asymptomatic (*p* = 0.041) and controls (*p* = 0.043); s = seconds.

**Table 5 jfmk-10-00414-t005:** Sensorimotor testing data between groups. Data reported in mean (±SD).

	Choice RT ^+^ (ms)	Choice RT Error (% Correct)	Sequential Amplitude Discrimination (μm)	Simultaneous Amplitude Discrimination (μm)	Temporal Order Judgement (ms)	Duration Discrimination (ms)
Symptomatic (n = 15; 12 m, 3 f)	467.0 (±118.8)	94.0 (±5.5)	49.2 (±37.3)	80.5 (±46.4)	42.1 ^1^ (±25.2)	65.3 (±38.5)
Asymptomatic (n = 10; 9 m, 1 f)	448.3 (±84.7)	90.0 (±5.8)	37.4 (±29.2)	54.5 (±23.3)	25.9 (±15.4)	64.2 (±46.3)
Controls (n = 20; 17 m; 3 f)	423.3 (±82.1)	93.0 (±4.8)	43.5 (±33.5)	55.0 (±34.5)	25.8 (±11.3)	53.5 (±20.1)

^1^  *p* = 0.038 symptomatic versus control; ^+^ RT = reaction time.

**Table 6 jfmk-10-00414-t006:** Transcranial magnetic stimulation. Data reported in mean (±SD).

	aMT * (% Maximal Stimulator Output)	Corticospinal Latency (ms)	cSP:MEP ^+^ Ratio			SICI ^#^ Ratio	LICI ^#^ Ratio
			130% aMT	150% aMT	170% aMT		
Symptomatic (n = 15; 12 m, 3 f)	35.6 (±8.2)	94.0 (±5.5)	49.3 (±24.3)	53.0 ^1^ (±20.4)	57.1 ^2^ (±19.6)	47.4 ^3^ (±9.4)	55.7 (±15.2)
Asymptomatic (n = 10; 9 m, 1 f)	34.0 (±5.7)	90.0 (±5.8)	40.1 (±20.7)	40.4 (±20.2)	41.3 (±22.9)	65.1 (±16.1)	68.5 (±13.3)
Controls (n = 20; 17 m; 3 f)	34.2 (±7.4)	93.0 (±4.8)	36.4 (±17.2)	30.6 (±14.5)	35.1 (±22.6)	76.2 (±11.3)	69.6 (±17.3)

^1^  *p* = 0.018 symptomatic versus control; ^2^
*p* = 0.046 symptomatic versus control; ^3^ *p* = 0.013 symptomatic versus asymptomatic and *p* = 0.001 symptomatic versus control; * aMT = active motor threshold; ^+^ cSP:MEP = cortical silent period–motor evoked potential; ^#^ SICI = short intracortical inhibition, LICI = long intracortical inhibition.

## Data Availability

Data are available upon reasonable request from an authorised institution.

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
