# Peer review of "Long-Term Cumulative Effects of Repeated Concussions in Cyclists: A Neurophysiological and Sensorimotor Study"

_jfmk, 2025, doi:10.3390/jfmk10040414_

Round 1
Reviewer 1 Report
Comments and Suggestions for Authors
The manuscript is well written, with good scientific rigour and clear methodological design. The results are well presented both in table, text and figures. The discussion is well reasoned within the findings of the study and the wider literature, and the limitations of the study are acknowledged. Some aspects of the writing can be improved for better clarity, please see below:
Abstract ln 16-19: The sentence is quite long and unclear. I would recommend rephrasing the content.
Introduction ln 88-90: The sentence is somewhat ambiguous as it implies that the parameters mentioned after the comma should be different from those in the first half of the sentence. I imagine that the authors meant that although both cSP and LICI reflect GABA_B receptor activity, these two parameters relate to different neural and circuit mechanisms? If so I would recommend rephrasing the sentence.
Introduction ln 95: Suggest changing "has supported" to "have supported"
Materials and Methods ln 120-126: It seems unusual to divide participants in symptomatic and asymptomatic based on whether they are concerned or not about their cognitive health - someone may be symptomatic but not be concerned, or viceversa. Could the authors clarify their decision to use this approach? As per table 2 the symptomatic participants do indeed report significantly higher symptoms than the asymptomatic group so this would be a more objective way of separating the groups.
Statistical analysis: From my understanding reaction time was analysed using a 2 Way ANOVA to account for the repeated tests, if this is the case it should be mentioned in this section
Results ln 291-292: Repetition of 'illustrates' and 'illustrating' - consider rephrasing.
Conclusion: This is a good summary but could draw more direct links between GABA dysfunction and vestibular/motor issues (as previously mentioned in the discussion).
Author Response
Reviewer 1
The manuscript is well written, with good scientific rigour and clear methodological design. The results are well presented both in table, text and figures. The discussion is well reasoned within the findings of the study and the wider literature, and the limitations of the study are acknowledged. Some aspects of the writing can be improved for better clarity, please see below:
Response: We thank Reviewer 1 for their feedback. We have addressed each comment below and in the manuscript.
Abstract ln 16-19: The sentence is quite long and unclear. I would recommend rephrasing the content.
Response: We agree that this sentence is overly long. We have broken this long sentence into three sentences now to clarify the aim of the study (lines 16-20).
Introduction ln 88-90: The sentence is somewhat ambiguous as it implies that the parameters mentioned after the comma should be different from those in the first half of the sentence. I imagine that the authors meant that although both cSP and LICI reflect GABA_B receptor activity, these two parameters relate to different neural and circuit mechanisms? If so I would recommend rephrasing the sentence.
Response: The reviewer is correct and agree that the sentence is confusing. As Reviewer 2 suggested a reduction in wording, we have rewritten the TMS section but have kept the sentence. We have removed the comma to reduce the ambiguousness and have reworded the latter half of the sentence to reflect the Reviewer’s suggestion (lines 101-102).
Introduction ln 95: Suggest changing "has supported" to "have supported"
Response: Have change wording from “has” to “have” (line 110).
Materials and Methods ln 120-126: It seems unusual to divide participants in symptomatic and asymptomatic based on whether they are concerned or not about their cognitive health - someone may be symptomatic but not be concerned, or viceversa. Could the authors clarify their decision to use this approach? As per table 2 the symptomatic participants do indeed report significantly higher symptoms than the asymptomatic group so this would be a more objective way of separating the groups.
Response: This is a good point raised by the Reviewer and one that has always been difficult to explain, given that in longitudinal concussion research a question has always been why some individuals continue to struggle, whilst others are apparently unaffected. We have taken the suggestion regarding a more objective way of separating the groups by using the Johansson et al (2009) clinical cut off criteria (lines 142-144).
Statistical analysis: From my understanding reaction time was analysed using a 2 Way ANOVA to account for the repeated tests, if this is the case it should be mentioned in this section
Response: Apologies for the typographical error. We have removed one-way ANOVA as a general overview in section 2.5, and now indicated one- or two-way ANOVAs where appropriate in the results section.
Results ln 291-292: Repetition of 'illustrates' and 'illustrating' - consider rephrasing.
Response: We have changed ‘illustrates’ to ‘presents’ (line 369).
Conclusion: This is a good summary but could draw more direct links between GABA dysfunction and vestibular/motor issues (as previously mentioned in the discussion).
Response: We thank the Reviewer for this and have incorporated this suggestion in the new Conclusion paragraph at the end of the manuscript (lines 470-474).
Reviewer 2 Report
Comments and Suggestions for Authors
comment in the attachment

Author Response
Reviewer 2
- Acknowledgment and importance of the topic
Thank you for the opportunity to review this article. The topic is highly relevant and timely,
addressing long-term neurological effects of repeated concussions in cyclists. The issue is of
growing concern due to the increasing awareness of brain injuries in sports, including non-
contact disciplines such as cycling.
Suggested improvement:
Add a brief introductory statement emphasizing the broader clinical and public health
relevance, such as the economic and occupational consequences of recurrent concussions in
athletes.
Response: We have included an extra paragraph in the Introduction section that emphasises the broader clinical and public health relevance, including personal economic and occupational consequences of recurrent concussions in
athletes.
- Relevance and significance of the problem
The article addresses a significant and current research gap. Previous studies have mostly
focused on contact sports, so this study fills an important void in cycling-related concussion
research.
Suggested improvement:
Include the most recent epidemiological data on cycling-related concussions from 2023–2024
to strengthen the argument for the study’s timeliness.
Response: We have now included data from an epidemiological study by Fallon et al.
- Contribution to knowledge
The article provides valuable insight into the neurophysiological consequences of repetitive
concussion in cyclists. It introduces a novel integration of transcranial magnetic stimulation
(TMS) and sensorimotor testing, not previously applied in this specific athletic population.
Suggested improvement:
Clarify in the Discussion section how this study extends or differs from the authors’ earlier
works in contact-sport populations.
Response: With due respect we have provided how this novel study differs from other work. In the second paragraph of the Discussion section, we state that, compared to previous studies in contact athletes, this is the first study completed in cycling populations. Moreover, we state in the same paragraph that while the cycling cohorts would be considered healthy via clinical reference limits, more rigorous testing from TMS suggests that persistent abnormalities continue which may contribute to statistically significant differences between groups. We hope the Reviewer appreciates that we have already addressed this.
- Originality and practical usefulnessThe study is original and methodologically sound. The use of both cognitive-motor and neurophysiological measures makes it unique.
Suggested improvement:
In the Conclusions, highlight practical applications of the findings, such as how TMS or
sensorimotor testing could be integrated into post-concussion protocols for cycling
federations.
Response: We have added in a final sentence of the conclusions paragraph to reflect this suggestion.
- Title and keywords
The title is appropriate and accurately reflects the study content. Keywords are relevant and
not redundant.
Suggested improvement:
Consider revising the title to:
“Long-Term Cumulative Effects of Repeated Concussions in Cyclists: A Neurophysiological
and Sensorimotor Study.”
Response: This is a good suggestion by the Reviewer, and we have included the extra wording in the title.
- Abstract
The abstract is well structured, including background, methods, results, and conclusions.
Suggested improvement:
- In the Methods section, specify the study design as “cross-sectional between-group.”
- In the Results section, mention effect sizes, for example: “Effect sizes were large for
cortical inhibition (Cohen’s d > 1.0).”
Response: We have added in wording as suggested by the Reviewer.
- Introduction
The introduction clearly states the research purpose, provides definitions, and identifies
literature gaps. However, it could be more concise and better supported by systematic
reviews.
Suggested improvement:
Condense the introduction by about 10–15 percent, and add reference to a recent systematic
review or meta-analysis on TMS findings in concussion research.
Response: We have condensed the introduction section (despite the added paragraph requested by the Reviewer on emphasising the broader clinical and public health relevance, including personal economic and occupational consequences of recurrent concussions in athletes). Specifically, we have reduced the TMS explanation paragraph and added a reference on a systematic review (the only one to date) on TMS findings in concussion research.
- MethodsThe methodology is detailed and rigorous. Ethical approval and inclusion/exclusion criteria
are clearly stated. However, there are some missing methodological elements.
Suggested improvement:
- Add a simple flow chart or schematic diagram showing participant recruitment and
group allocation.
- Include a short statement on sample size estimation (for example, G*Power
calculation for ANOVA with effect size f=0.25, alpha=0.05, power=0.80).
Response: We have now included a new Figure 1 and a short statement on sample size estimation.
- Data analysis and conclusions
The statistical approach is correct, with use of ANOVA, post-hoc Tukey tests, and Cohen’s d
effect sizes. Conclusions are logical and data-supported.
Suggested improvement:
- Add a note on the clinical interpretation of significant findings (for example, reaction
time differences greater than 50 ms may be clinically meaningful).
- In the conclusion, emphasize how these methods can support return-to-sport decision-
making.
Response: We have added a note on the clinical interpretation of the findings (lines 416-420). We have also added a sentence in the conclusion (last sentence) to emphasize physiological testing can support cyclists following concussion as per the Reviewer’s comment #4.
- Discussion
The discussion is well written, beginning with a restatement of the aims and key results, and
comparing outcomes with previous studies. Limitations are mentioned but not structured.
Suggested improvement:
Create a distinct subsection titled “Strengths and Limitations,” listing both strong points
(innovative use of TMS and sensorimotor methods) and limitations (self-reported symptoms,
lack of musculoskeletal injury data).
Response: We have added in a strengths and limitations sub-section as per the Reviewer’s suggestion.
- Structure, figures, and references
The structure follows a logical IMRD format. Figures and tables are clear, but figure
captions could be more descriptive. References are recent and appropriate.
Suggested improvement:
- Expand figure captions with a short note on clinical interpretation.
- Add one or two recent citations from 2024 related to head trauma in cycling.
Response: We appreciate the Reviewer’s comment on figure legends; however, figure legends should not include clinical interpretations but rather provide an objective description of the figure or table. We have provided interpretations of the data throughout the discussion section for the Reader.
We have added some recent citations as per the Reviewer’s suggestions earlier in this reviewer’s report.
- Editorial and technical notes
- Ensure consistent use of abbreviations (SRC, RHI, TMS) at first appearance.
- Standardize the format of tables (mean ± SD or equivalent).
- Verify that all figures are correctly referenced in the text (for example, “as shown in
Figure 2”).
Response: All abbreviations, data format and referencing of figures and tables are correct.
Overall assessment
The article demonstrates a high scientific and methodological standard. It provides original
data and valuable insights into the neurophysiological effects of repeated concussions in
cyclists.
Final recommendation: Accept - minor revisions.